# Improving Behavioural Cloning with Positive Unlabeled Learning

**Qiang Wang[1], Robert McCarthy[2], David Cordova Bulens[1],
Francisco Roldan Sanchez[3,4], Kevin McGuinness[3,4], Noel E. O'Connor[3,4],
Nico Gürtler[5], Felix Widmaier[5], Stephen J. Redmond[†1,4]**

[1]University College Dublin, [2]University College London, [3]Dublin City University
[4]Insight SFI Research Centre for Data Analytics, [5]MPI for Intelligent Systems

**Abstract:** Learning control policies offline from pre-recorded datasets is a promising avenue for solving challenging real-world problems. However, available datasets are typically of mixed quality, with a limited number of the trajectories that we would consider as positive examples; i.e., high-quality demonstrations. Therefore, we propose a novel iterative learning algorithm for identifying expert trajectories in unlabeled mixed-quality robotics datasets given a minimal set of positive examples, surpassing existing algorithms in terms of accuracy. We show that applying behavioral cloning to the resulting filtered dataset outperforms several competitive offline reinforcement learning and imitation learning baselines. We perform experiments on a range of simulated locomotion tasks and on two challenging manipulation tasks on a real robotic system; in these experiments, our method showcases state-of-the-art performance. Our website: https://sites.google.com/view/offline-policy-learning-pubc.

**Keywords:** Offline policy learning, Positive unlabeled learning, Behavioural cloning

## 1 Introduction

Data-driven learning methods can discover sophisticated control strategies with minimal human involvement, and have demonstrated impressive performance in learning skills across many challenging domains [1, 2, 3, 4]. Nonetheless, data-driven methods are not often applied in real world applications due to the amount of interactions with the environment needed before an effective policy can be learned [5, 6, 7]. Moreover, data acquisition can be costly and/or unsafe in physical environments. This inefficiency can potentially be improved by learning from previously-collected data; i.e., learning a policy from a historical dataset without needing additional data acquisition from the environment. This is termed offline policy learning.

Standard behavioral cloning (BC) is the simplest offline policy learning algorithm, it aims to find a policy that can mimic the behavior observed in a dataset capturing the performance of a given task. The target behavior to be cloned is usually obtained from an expert; for instance, a human [8, 9] or a well-performing scripted agent [10]. BC performs supervised regression, learning a control policy that maps observations from a dataset to the corresponding actions taken by a behavior policy. When high-quality expert data is used for training, BC demonstrates high efficiency, and the resulting agent typically exhibits good performance [11].

However, a major drawback of BC is its dependence on a high-quality training dataset. Specifically, the data collected for BC should come from one highly-skilled expert. Moreover, the actions conditioned on the environment states must display a unimodal distribution to prevent regression ambiguity [12], as BC follows a supervised machine learning paradigm. However, real-world/practical training datasets are often of mixed-quality, containing examples of both high-quality and low-quality

---

† is the corresponding author

behaviors/data, which can be detrimental to the training process. The presence of low-quality data in these datasets can be attributed to factors such as the involvement of low-skilled agents, non-task-related policies, or environmental noise. If multiple experts are involved, the dataset may furthermore exhibit multi-modal behavior, leading to regression ambiguity in BC. In this paper, we aim to study methods allowing the discrimination of a single target expert's data within a mixed-quality dataset, and utilize this single-expert subset for BC learning.

## 1.1 Our work

We assume we are provided with a mixed-quality dataset, as discussed previously, along with a seed-positive dataset consisting of high-quality data generated by an expert. Our goal is to discriminate data in the mixed-quality dataset that shares the same behavioral patterns as the seed-positive examples to obtain an expert dataset suitable for BC. In practice, the seed-positive dataset can be quite small, typically constituting only 0.1% to 0.4% of the mixed-quality dataset's size. This enables users to obtain the seed-positive examples using feasible methods, such as heuristically/manually sampling from the mixed-quality raw dataset or by requesting the target expert to collect a small amount of additional data. Our approach can be considered an example of Positive Unlabeled (PU) learning, which constitutes an important subfield within conventional semi-supervised learning. Concisely, datasets for PU learning comprise a portion of data labeled as positive, while the rest remains unlabeled. The objective of PU learning is to leverage the information from positive examples to classify the unlabeled data as either positive or negative.

We establish a supervised learning signal by integrating synthetically-generated negative examples and seed-positive examples. Our training methodology adheres to a traditional semi-supervised learning paradigm; it begins the training with a small, positive dataset and then iteratively discriminates positive examples from a large, unlabeled dataset. These identified examples are then added to the positively-labeled subset for the subsequent training cycle, until convergence is achieved. Once the dataset has been labeled using this iterative method, during the policy learning phase, we apply standard BC to this positively-labeled subset.

In general, we refer to our offline policy learning approach as Positive Unlabeled Behavioural Cloning (PUBC). PUBC stands out for its simplicity in implementation, quick training time, and ease of parameter tuning. In our experiment, PUBC achieves excellent performance across a wide range of challenging robotic tasks, surpassing several state-of-the-art algorithms.

## 2 Related work

### 2.1 Offline policy learning

Research methods for offline policy learning can largely be categorized as offline imitation learning (IL) or offline reinforcement learning (RL); we refer readers to [13, 14] for comprehensive surveys, and to [15, 16, 17] for research benchmarks.

BC is the simplest form of offline IL. In addition to standard BC as mentioned earlier, researchers have recently proposed Implicit BC [18], which employs an energy-based model [19] to improve BC's performance. Furthermore, Inverse RL [20] is regarded as an alternative branch of IL, focusing on understanding the motivations behind an agent's actions by inferring the underlying reward structure, which guides the decision-making process in Inverse RL. Additionally, a Generative Adversarial Network (GAN) [21] has been integrated with IL in [22], where it employs adversarial training with generative and discriminative models to learn the action distribution of the behavior policy.

Offline RL[14] aims to maximize the expectation of the sum of discounted rewards. However, unlike in online RL, no interactions with the environment are allowed. Most off-policy RL [23] algorithms are applicable offline, but they typically suffer from the issue of outputting out-of-distribution (OOD) actions due to the distribution shift between the action distribution in the training dataset and that induced by the learned policy [10]. To mitigate this issue, several constraint methods, such as policy regularization [24, 10, 25, 26] and conservative value estimates [27, 28] are proposed.

Recently, a novel offline RL approach, inspired by transformer models [29], has been introduced in [30, 31]. Unlike conventional RL methods that depend on policy gradients or temporal differences, this approach takes a distinct route by utilising supervised sequence modelling to fit the policy data distribution. In the policy evaluation phase, the GPT model takes a target return and generates an action sequence for that goal. This GPT-based method will be a baseline in our paper.

Another branch of offline RL perfroms BC while focusing on learning transitions with higher significance in the datasets. It starts by learning advantage functions and subsequently utilizes them to downweight transitions with lower advantage [32, 33, 34, 35, 36, 37]. Similarly, in [38], the authors propose a GAN-like architecture [21] that combines BC and a discriminator. The discriminator selectively picks high-quality expert data from the dataset for BC learning.

### 2.2 Positive unlabeled learning

The main challenge in PU learning is to acquire negative examples from the unlabeled data to complement the available positive examples for training a supervised classifier. The two-step PU learning approach involves manually labeling a subset of negative examples and using them along with positive examples to train the classifier [39]. The classifier can then label the remaining unlabeled data in the dataset. However, this method can be human labor-intensive, and it may not be effective if the underlying patterns of the negative examples are difficult to interpret.

Another solution is to naively treat the unlabeled data as negative examples during classifier training [40]. The classifier can then assign scores to the unlabeled examples, with positive examples typically receiving higher scores. This method has been improved in [41] by using the bagging technique to generate multiple subsets from the unlabeled dataset, which are combined with positive examples to train a series of weaker classifiers. Finally, the output from the classifier ensemble is used to produce a more accurate prediction. However, if an unsuitable loss function is used, biased errors may occur. This problem is addressed in [42] by introducing Unbiased PU learning. More recently, [43] proposed Non-negative PU learning to mitigate the overfitting problem associated with unbiased PU learning. We recommend [44] for an in-depth analysis of PU learning.

## 3 Positive Unlabeled Behavioural Cloning

### 3.1 Preliminaries

The offline policy learning problem is formulated in the context of a Markov decision process, $\mathcal{M}$ = $(\mathcal{S}, \mathcal{A}, \mathcal{R}, \mathcal{P}, \gamma)$ [45], where $\mathcal{S}$ is the state space, $\mathcal{A}$ is the action space, $\mathcal{R}$ is the reward function, $\mathcal{P}$ is environment dynamic and $\gamma$ is the discount factor. At each time step, $t$, the agent gets a state $s_t \in \mathcal{S}$ and outputs an action $a_t \in \mathcal{A}$ according to a policy $\pi(a_t \mid s_t)$; after applying the action to the environment, the agent will get a reward $r_t \in \mathcal{R}$ and the environment state transitions to $s_{t+1}$. We assume that we can obtain or are given a positive dataset, $\mathcal{D}_+ = \{(s_t^+, a_t^+, r_t^+, s_{t+1}^+)_{t=1...m}\}$, with $m$ time steps and a large mixed-quality offline dataset $\mathcal{D}_{mix} = \{(s_t^{mix}, a_t^{mix}, r_t^{mix}, s_{t+1}^{mix})_{t=1...n}\}$, with $n$ time steps. We assume $m << n$, that $\mathcal{D}_+$ contains only data collected by the targeted expert, and $\mathcal{D}_{mix}$ includes a proportion of data collected by the target expert. In our following description, we define a positive example as the data collected by the target expert, and a negative example as any data not collected by the target expert.

### 3.2 Generating the training examples

Our approach to generating negative examples for training the PU classifier is similar to the two-step method outlined in Section 2.2. However, rather than depending on manual selection of negative examples from $\mathcal{D}_{mix}$, we create them by randomly mixing the states and actions from different sources, including from $\mathcal{D}_+$, from $\mathcal{D}_{mix}$, and random examples from the state-action space (see Figure 1(a)). Therefore, the set of negative examples can be informally written as: $\mathcal{D}_- = \{(s^+, a^{mix})_{n_1} \cup (s^+, \tilde{a})_{n_2} \cup (s^{mix}, a^+)_{n_3} \cup (s^{mix}, \tilde{a})_{n_4} \cup (\tilde{s}, a^+)_{n_5} \cup (\tilde{s}, a^{mix})_{n_6} \cup (\tilde{s}, \tilde{a})_{n_7}\}$, where $\tilde{s}$ and $\tilde{a}$ refer to the random states and actions that follow a uniform distribution within the range of minimum action to maximum action; $n_1 - n_7$ correspond to the number of state-action pairs generated for each combination of sources. We require that the artificially generated state-action pair be distinct from the state-action pair obtained from the raw dataset.

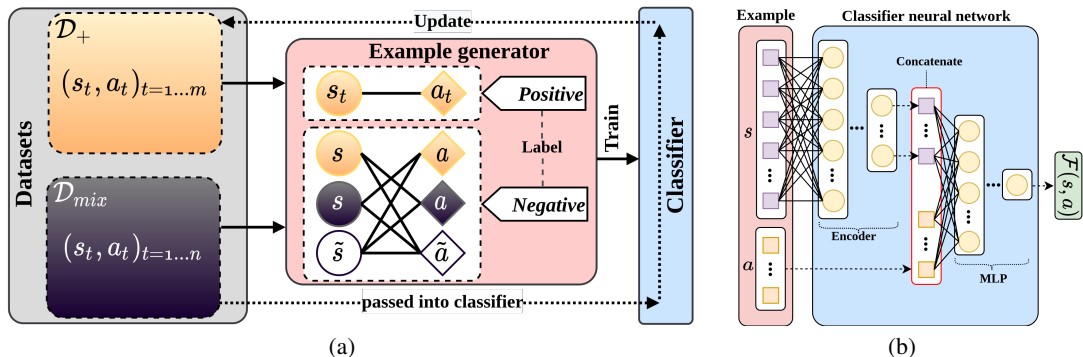

(a)                                          (b)

Figure 1: PUBC learning block diagram. (a) Illustration of the iterative PU learning process. The red rounded rectangle shows how *positive* and *negative* training examples are generated, with the *negative* examples generated by intentionally mismatching actions and states from different time points and/or different data subsets to form action-state pairs that would most likely represent poorly performing behaviors; See section 3.2 for a more comprehensive explanation. (b) The neural network structure of the classifier includes an encoder to reduce the usually high-dimensional state vector, $\int$, followed by a multilayer perceptron (MLP) classifier, taking as input both state $\int$ and action $\mathbf{a}$ vectors, and whose sigmoidal output is interpreted as the probability that the example was generated by an expert agent.

In the first training iteration, we treat the examples in the positive dataset as positive, similar to traditional PU learning. Once the classifier is trained, we use it to identify additional positive examples from the unlabeled dataset. These newly identified positive examples then replace the previous ones.

### 3.3 Classifier structure

Figure 1(b) shows the network structure of the classifier, which is a combination of MLPs; it takes the state-action pair and outputs the probability that this pair was generated by the actions of the same expert agent that generated (we assume) most of the data in the training dataset, $\mathcal{D}_+$. Experimentally, we found that directly inputting a concatenation of raw state-action pairs into the neural network will cause the action to be effectively ignored, as the dimension of the observation space is typically much larger than the action space in practical task settings. Therefore, to reduce the dimension of the observation space, we encode the observation vector into a lower dimensional space before concatenating it with the action vector and inputting it to the MLP to obtain a decision. The final layer of the MLP uses a sigmoid activation function, $\text{sigmoid}(x) = 1/(1 + e^{-x})$, where the output $\mathcal{F}(s, a)$ is interpreted as the probability that the state-action pair was generated by an expert. Binary cross-entropy is chosen as the training loss, $\mathcal{L}$:

$$\mathcal{L} = \mathop{\mathbb{E}}_{(s_t,a_t) \sim \mathcal{D}_-} \left[ -\log(1 - \mathcal{F}(s_t, a_t)) \right] + \mathop{\mathbb{E}}_{(s_t,a_t) \sim \mathcal{D}_+} \left[ -\log \mathcal{F}(s_t, a_t) \right]. \tag{1}$$

### 3.4 Additional methods for optimizing PU learning performance

- **Classifying per trajectory:** The policy used for data collection typically does not change within a trajectory (each interaction episode). The labels assigned to all state-action pairs in a trajectory are therefore aggregated and the same label is applied to all transitions in the trajectory. This is done by soft voting, taking the mean of the individually predicted probabilities of the state-action pairs in the trajectory, $\mathcal{F}(s_t, a_t)$ for $t$ a member of the trajectory time interval, to create a *confidence score* for the trajectory. Subsequently, a threshold, $th_{\text{conf}}$, is applied to the confidence score to binarize it.

- **Adaptive confidence threshold:** It is necessary to set a confidence threshold $th_{\text{conf}}$ to discriminate positive trajectories in the unlabeled dataset, $\mathcal{D}_{mix}$. In other words, all trajectories with a confidence score exceeding the threshold are classified as subsets containing entirely positive examples. We discovered that employing an adaptive threshold can enhance the classifier's performance. Briefly, it searches for a local minimum in the confidence score histogram and uses this value as a decision

---

**Algorithm 1:** PUBC algorithm

---

**Input:** Mixed-quality dataset $\mathcal{D}_{mix}$, positive dataset $\mathcal{D}_+$

**while** $\mathcal{D}_+$ *not converged* **do**

    Randomly sample $K$ subsets $\{\mathcal{D}_+^k\}_{k=1...K}$ from $\mathcal{D}_+$;

    Generate $K$ corresponding *negative* subsets $\{\mathcal{D}_-^k\}_{k=1...K}$;

    Initialise $K$ classifiers $\{\mathcal{F}_{\theta_k}\}_{k=1...K}$ with parameters $\theta_1, ..., \theta_K$;

    **for** $k \leftarrow 1$ **to** $K$ **do**

        | Update $\theta_k$ by minimizing the loss in Equation 1;

    **end**

    Get *confidence* threshold $th_{\text{conf}}$ using the trained $\{\mathcal{F}_{\theta_k}\}_{k=1...K}$;

    Update the membership of $\mathcal{D}_+$ using Equation 2;

**end**

Initialise BC network $\pi_\delta$ with parameters of $\delta$;

**for** $epoch \leftarrow 1$ **to** $epochs$ **do**

    Update $\delta$ by minimizing: $\underset{(s_t, a_t) \sim \mathcal{D}_+}{\mathbb{E}} \left[ -\log \pi_\delta(a_t \mid s_t) \right]$

**end**

**Output:** $\pi_\delta$

---

boundary, above which it is assumed the data is dominated by expert-generated trajectories (see Appendix B for details).

- **Ensemble learning:** Similar to the approach in [41], we employ bagging to enhance the classifier's performance by concurrently learning multiple independent weaker classifiers and combining their individual decisions to determine a final decision. The datasets used to train each classifier are subsets of $\mathcal{D}_+$, sampled with replacement.

In Appendix D, we conducted an ablation study on these three techniques individually.

### 3.5 Using the trained classifier and learning the policy

Considering the above techniques, the label decision for a trajectory can be determined using:

$$f := \mathbb{1}\left[ \sum_{k=1}^{K} \mathbb{1}\left[ \left( \frac{1}{T} \sum_{t=1}^{T} \mathcal{F}_k(s_t, a_t) \right) \geq th_{\text{conf}} \right] > \frac{K}{2} \right], \tag{2}$$

where $\mathbb{1}[\cdot]$ are indicator functions, $T$ is the number of time steps in each trajectory; and $K$ is the number of classifiers in the ensemble, set to an odd number to avoid ties.

The process iterates until trajectories in $\mathcal{D}_+$ converge. Once $\mathcal{D}_+$ has converged, we train a standard BC model to mimic the behavior in $\mathcal{D}_+$. The entire semi-supervised training and filtering process for our PU classifier as well as the training of BC can be succinctly summarized in Algorithm 1.

## 4 Experimental results

This section aims to showcase the effectiveness of our proposed PUBC method by conducting experiments on a range of continuous control benchmark tasks, including challenging physical robotic manipulation tasks from the Real Robot Challenge (RRC) III competition[1] and numerous MuJoCo locomotion tasks [47, 48] (see Figure 2).

### 4.1 Environments, tasks and datasets

#### 4.1.1 Robotic manipulation tasks

Figure 2(a) illustrates the domains of RRC III, which focuses on two tasks: **Push** and **Lift** [49]. In the **Push** task, the cube must be moved to specified positions on the arena floor. The more challenging **Lift** task requires lifting the cube and maintaining it at a target position and orientation.

---

[1]A robotic manipulation competition featured in the NeurIPS 2022 Competition Track, more details see https://real-robot-challenge.com/. We won the competition [46], with the filter-based technique being one of our key strategies, and the proposed PUBC method extends this filtering method.

In our experiment, we use the mixed datasets provided by the competition organizers for each task. Each mixed dataset is collected by a mixture of different policies with varying levels of skill, and a significant portion of the data was collected by domain-specific experts, ensuring high quality. The subsequent discussion will denote these datasets by **Lift/mixed** and **Push/mixed**, respectively.

To obtain the seed-positive dataset, $\mathcal{D}_+$, we assume that the target expert has generated the trajectories with the highest returns. Therefore, we select the 0.4% episodes with the highest returns as our seed-positive dataset. The selection of this value is further investigated in Appendix D through an ablation study. Post-competition, we received trajectory labels from RRC III organizers, serving as ground truth to assess our PU learning method's accuracy.

### 4.1.2 MuJoCo locomotion tasks

As shown in Figure 2(b)-2(c), the bodies being controlled in the locomotion tasks comprise segments and joints. Actions are applied to maintain the balance of the body and to move forward. Here, We collected mixed-quality datasets comprising five different structures: 1. **E+E**: Expert+Expert dataset consisting of data from two expert policies with similar performance but different habits/behaviors, with only one expert policy being of interest to us; 2. **E+W**: Expert+Weaker dataset containing data from one expert policy and one poorly performing agent; 3. **E+N**: Expert+Noise dataset comprising expert data and an equal amount of domain noise; 4. **E+E+W+N**: a combination of the above four types of data. In addition, we also included an expert dataset **E**, consisting only of expert data as a baseline. The configuration of each mixed dataset is further detailed in Appendix C.

## 4.2 PU learning results

This section presents the accuracy of our PU learning method, comparing it to traditional PU learning approaches, including Unbiased PU learning and Non-negative (NN) PU learning. The mathematical formulations for Unbiased PU and NN PU learning methodologies are provided in Appendix F.1 for reference. To maintain a balanced comparison, we also employed the techniques delineated in Section 3.4 to the baselines.

As illustrated in Table 1, there is no significant difference between the performances of the two baseline methods, while our approach shows a significant increase in accuracy compared to these methods. The baseline methods. This could potentially be attributed to the relatively complex data distribution produced by the RRC III environment. In locomotion tasks, both the environment and the task are relatively stable, typically involving the operation of a simulated body to complete the single task of moving forward. However, in the RRC III manipulation environment, there is a high degree of randomness; both the initial position of the object and the target are randomly initialized, meaning the tasks completed in each trajectory vary. This results in a more complex data distribution. Furthermore, we observed that generally, in real-world environments, the performance of traditional algorithms tends to be inferior to their performance in simulators. This is due to non-ideal hardware in physical environments introducing substantial environmental noise, making the data distribution even more complex.

In unbiased PU learning, introducing a reweighting operation to mitigate positive label data bias can shift the data distribution, resulting in poor model generalization, especially in complex data distributions like the RRC III environment.

In the Non-negative PU learning, the weight of the positive samples are enforced to non-negative, however, this may hinder the model from capturing essential nuances in the data. For some intricate data distributions, permitting the model to allocate negative weights to positive samples could be instrumental in uncovering the data's inherent structure and relationships more effectively.

In contrast to this, the method we propose can effectively tackle these issues, demonstrating both high accuracy and strong robustness.

## 4.3 Offline policy learning

We present the evaluated policy performance in Table 2, where we compare our PUBC with other relevant baseline algorithms, including a naive reward-based filter before performing BC, CRR[32],

Table 1: Comparing our method's accuracy to baselines in classifying expert vs. non-expert trajectories. Accuracy formula: $(TP + TN)/(TP + TN + FP + FN)$, where $TP$ (True Positive) and $TN$ (True Negative) denote the correct classification of expert trajectories and non-expert trajectories respectively. $FP$ (False Positive) denotes that non-expert trajectories are incorrectly classified as expert, $FN$ (False Negative) denotes that expert trajectories are incorrectly classified as non-expert.

| Dataset | Unbiased PU | NN-PU | Ours |
|---|---|---|---|
| RRC-Sim-**Lift/mixed** | 82.5% | 79.2% | 99.7% |
| RRC-Sim-**Push/mixed** | 94.3% | 90.5% | 100.0% |
| RRC-Sim Avg | 88.4% | 84.9% | **99.9%** |
| RRC-Real-**Lift/mixed** | 69.4% | 64.8% | 99.2% |
| RRC-Real-**Push/mixed** | 88.7% | 90.0% | 100.0% |
| RRC-Real Avg | 79.1% | 77.4% | **99.6%** |
| Ant - **E+E** | 98.1% | 95.3% | 99.0% |
| Ant - **E+W** | 100.0% | 99.2% | 100.0% |
| Ant - **E+N** | 99.3% | 100.0% | 100.0% |
| Ant - **E+E+W+N** | 98.1% | 96.9% | 98.8% |
| Ant Avg | 98.9% | 97.9% | **99.5%** |
| Humanoid - **E+E** | 93.4% | 94.9% | 99.7% |
| Humanoid - **E+W** | 93.7% | 94.6% | 100% |
| Humanoid - **E+N** | 92.1% | 93.6% | 100% |
| Humanoid - **E+E+W+N** | 91.0% | 89.7% | 99.0% |
| Humanoid Avg | 92.6% | 93.2% | **99.7%** |
| Overall Avg | 89.7% | 88.3% | **99.6%** |

DWBC[38] and IQL[24]. We provide a detailed description of these baselines in Appendix F.2 for reference. It is evident that our PUBC consistently outperforms the baseline approaches across all domains. From the overall scores, we can observe that our method demonstrates a performance advantage of over 12% compared to the second-best performing approach.

Employing naive reward-based techniques like 10% BC and 50% BC can extract sufficient expert data from mixed-quality datasets in some cases. This is evident from the performance on the Sim- and Real-Push/mixed datasets, and the E+W dataset for both locomotion tasks. However, naive filtering works well only if there's a large performance difference between expert and non-expert policies, the fraction of expert data is known, and reward noise is low. Therefore, BC performs poorly on other datasets lacking these conditions.

The advantage-based CRR algorithm performs relatively poorly. This ineffectiveness stems from the challenging nature of estimating advantages in the offline RL setting, particularly when the environment is stochastic and/or the given rewards are sparse or noisy. In contrast, by considering behaviors over an entire trajectory, our method more accurately identifies target expert trajectories.

While IQL is considered one of the top-performing offline RL algorithms, our experiments have shown that its performance is not satisfactory on the selected problems. Although offline RL theoretically has the ability to handle various types of data, including mixed-quality datasets, it is preferable to use high-quality datasets in any scenario. Previous research has demonstrated that offline RL algorithms generally struggle to handle suboptimal robotics data effectively [50]. Furthermore, the additional experiments in Appendix E show that our PU method can improve the performance of offline RL on a range of D4RL benchmarks by enhancing the quality of the training data.

The GPT-based policy generation method DT, demonstrates strong performance in most RRC tasks, but it falls short in the Real Lift/Mixed tasks. We have noticed that DT requires about 6ms to generate an action at each time step. However, considering the demanding dexterity requirements of the Real Lift task, it can only tolerate a maximum delay of 2ms. This computational delay presents a limitation that would hinder the practical extension of DT into real-world scenarios. Another constraint is its reliance on high-quality data; its effectiveness reduces with noisy datasets, though it performs robustly on the expert's locomotion tasks.

Table 2: Averaged normalized scores of our method and the baselines. Each result is averaged over three training seeds, and training lasts $10^6$ time steps. We evaluate each learned policy for 100 trajectories. The score is normalized by $score_{norm} = (score - score_{min})/(score_{max} - score_{min})$.

| Dataset | Data | 10% BC | 50% BC | BC | DT | IQL | CRR | DWBC | PUBC (Ours) |
|---|---|---|---|---|---|---|---|---|---|
| RRC-Sim-**Lift/mixed** | 0.83 | 0.39 | 0.40 | 0.56 | 0.73 | 0.63 | 0.49 | 0.71 | **0.87** |
| RRC-Sim-**Push/mixed** | 0.61 | 0.64 | 0.84 | 0.59 | 0.80 | 0.71 | 0.82 | 0.80 | **0.85** |
| RRC-Sim total | 1.44 | 1.02 | 1.24 | 1.15 | 1.53 | 1.34 | 1.32 | 1.51 | **1.72** |
| RRC-Real-**Lift/mixed** | 0.60 | 0.32 | 0.31 | 0.31 | 0.44 | 0.36 | 0.40 | 0.54 | **0.65** |
| RRC-Real-**Push/mixed** | 0.44 | 0.67 | **0.85** | 0.58 | 0.82 | 0.79 | 0.79 | 0.81 | 0.83 |
| RRC-Real total | 1.04 | 1.00 | 1.15 | 0.89 | 1.26 | 1.15 | 1.20 | 1.36 | **1.49** |
| Ant - **E** | 0.80 | - | - | 0.84 | 0.87 | 0.82 | 0.76 | - | - |
| Ant - **E+E** | 0.78 | 0.70 | 0.68 | 0.73 | **0.84** | 0.80 | 0.68 | 0.71 | 0.79 |
| Ant - **E+W** | 0.53 | **0.82** | 0.80 | 0.53 | 0.73 | 0.79 | 0.41 | 0.80 | 0.79 |
| Ant - **E+N** | 0.42 | 0.62 | 0.54 | 0.47 | 0.52 | 0.74 | 0.73 | 0.76 | **0.79** |
| Ant - **E+E+W+N** | 0.48 | 0.70 | 0.53 | 0.31 | 0.67 | 0.67 | 0.29 | 0.69 | **0.78** |
| Ant total | 2.22 | 2.84 | 2.56 | 2.03 | 2.76 | 3.00 | 2.12 | 2.95 | **3.16** |
| Humanoid - **E** | 0.92 | - | - | 0.87 | 0.91 | 0.78 | 0.23 | - | - |
| Humanoid - **E+E** | 0.90 | 0.34 | 0.73 | 0.69 | **0.88** | 0.72 | 0.26 | 0.69 | 0.85 |
| Humanoid - **E+W** | 0.58 | 0.87 | **0.92** | 0.29 | 0.83 | 0.45 | 0.46 | 0.78 | 0.86 |
| Humanoid - **E+N** | 0.63 | 0.45 | 0.37 | 0.25 | 0.59 | 0.11 | 0.60 | 0.81 | **0.86** |
| Humanoid - **E+E+W+N** | 0.65 | 0.57 | 0.45 | 0.25 | 0.72 | 0.29 | 0.53 | 0.60 | **0.82** |
| Humanoid total | 2.76 | 2.23 | 2.48 | 1.47 | 3.02 | 1.57 | 1.85 | 2.88 | **3.40** |
| Overall | 7.46 | 7.09 | 7.43 | 5.55 | 8.57 | 7.06 | 6.48 | 8.70 | **9.76** |

DWBC demonstrates the second-best overall performance in our experiment. Nonetheless, its efficacy is notably limited in challenging RRC Sim- and Real- **Lift/mixed** datasets. The filtering/weighting component of DWBC shares similarities with classical PU learning methods. However, these approaches have limitations when dealing with complex data distributions, as mentioned previously.

## 5 Discussion

In summary, our PUBC method demonstrates superior accuracy and stability compared to conventional approaches in filtering the expert trajectories from the mixed-quality datasets. Furthermore, our PUBC can effectively learn policies from mixed-quality continuous control datasets, outperforming a variety of sophisticated state-of-the-art algorithms.

For certain applications, annotating rewards for all transitions can be a costly endeavor, especially in complex, real-world scenarios. Therefore, an alternative approach is to leverage our method to extract high-performing trajectories from a mixed-quality dataset by incorporating a few demonstration samples instead of manually crafting a reward function.

Of course, our methodology has certain limitations that should be acknowledged. Firstly, it is not applicable when the initial seed-positive dataset is unattainable. Secondly, when the data is not group in trajectories but consists of disorganized transitions, the performance of PUBC will be harmed. Lastly, our policy learning algorithm BC is inherently upper bounded by the performance of the expert behavior policy. Indeed, sophisticated RL related algorithms can often learn policies that generalize better and are able to directly learn from unknown datasets. In our future endeavors, we strive to enhance the performance of the trained policy by integrating the principles of our PU learning method with subsequent RL paradigms. Furthermore, we have plans to tackle more complex scenarios, including environments with partial observability and non-Markovian dynamics.

## 6 Conclusion

This paper introduces a new offline policy learning method termed PUBC, an effective approach to identifying transition behaviors generated by a specific expert policy in order to improve the quality of the dataset used for subsequent offline policy learning. This approach allows a learning algorithm to disregard low-skill behaviours, hence improving the performance of the learned policy. In our work, the PU learning method allows a naive BC learning algorithm to outperform other state-of-the-art offline RL algorithms in challenging physical problem domains.

**Acknowledgments**

This publication has emanated from research conducted with the financial support of China Scholarship Council under grant number CSC202006540003 and of Science Foundation Ireland under grant numbers 17/FRL/4832 and SFI/12/RC/2289_P2. We are grateful about the invaluable suggestions and comments that reviewers given to help to imporve the quality of this paper. We extend our heartfelt gratitude to Dr. Kevin McGuinness for his invaluable contributions and expertise to this research. It is with deep sorrow that we note he did not live to see the completion of this work. His exceptional insights and unwavering dedication will forever be etched in our memories.

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

## A  Environments

The environments of the tasks considered in our work are illustrated in Figure 2.

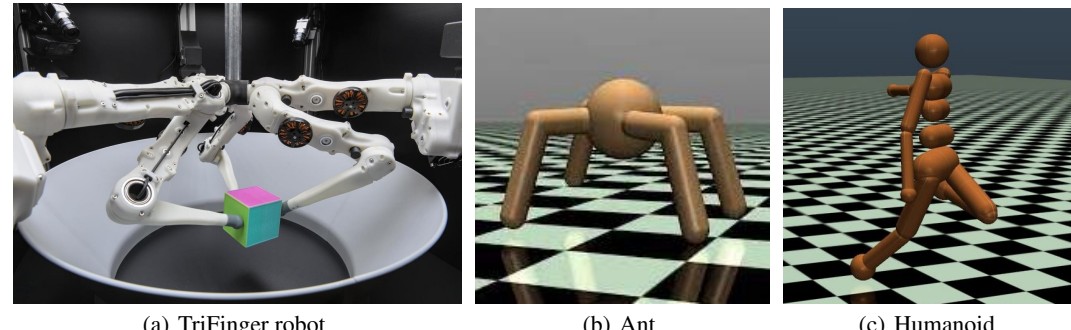

|          (a) TriFinger robot          |          (b) Ant          |          (c) Humanoid          |

Figure 2: (a) The physical TriFinger robot from the RRC III competition, where three identical robotic fingers are equally spaced 120° apart around the circular arena; the coloured cube is the object to be moved. (c)-(b) Illustrate the MuJoCo locomotion task environments.

## B  Description of how the adaptive confidence threshold is set

We use an adaptive mechanism to adjust the confidence threshold, $th_{\text{conf}}$, which is used in Section 3.5 to convert the continuous classifier output probability to a discrete binary label. Figure 3 shows an example of selecting the $th_{\text{conf}}$ value over the PU training iterations for the RRC III competition **Push/mixed** and **Lift/mixed** datasets. Furthermore, we presented the filtering accuracy across iterations for these two examples to better depict the convergence of accuracy.

We firstly apply the trained classifier to $\mathcal{D}_{mix}$ to get confidence scores (a probability, as the final layer activation function is sigmoidal) that each trajectory was generated by an expert policy. Secondly, we calculate the histogram of these confidence scores across all trajectories. Finally, we use a polynomial to fit the confidence score histogram. The threshold is determined by identifying the confidence score at which the maximum local minimum on the x-axis occurs. This point is marked with a blue dot in the bottom row of subplots. As the iterative training process proceeds, the trajectories selected by the PU learning changes and eventually the filter output converges. In our work, we define convergence as the condition where the change in trajectory memberships between two consecutive iterations is within a threshold of 2%.

Table 3: Illustration of the convergence process of filtering the RRC **Lift/mixed** and **Push/mixed** dataset. Each iteration lasts for 20 epochs. The TP (True Positive) represents an expert-collected trajectories that are correctly classified as expert-collected. FP (False Positive) represents a non-expert-collected trajectories that are incorrectly classified as expert-collected. FN (False Negative) represents expert-collected trajectories that are incorrectly classified as non-expert-collected, and TN (True Negative) represents non-expert-collected trajectories that are correctly classified as non-expert-collected.

|            | Iteration | TP   | FP | FN  | TN   |
|------------|-----------|------|----|-----|------|
|            | 1         | 200  | 4  | 998 | 1193 |
| **Lift/mixed** | 2     | 1005 | 97 | 193 | 1100 |
|            | 3         | 1195 | 37 | 3   | 1160 |
|            | 4         | 1194 | 9  | 4   | 1188 |
|            | 1         | 1421 | 2  | 429 | 1918 |
| **Push/mixed** | 2     | 1915 | 0  | 5   | 1100 |
|            | 3         | 1920 | 0  | 0   | 1920 |

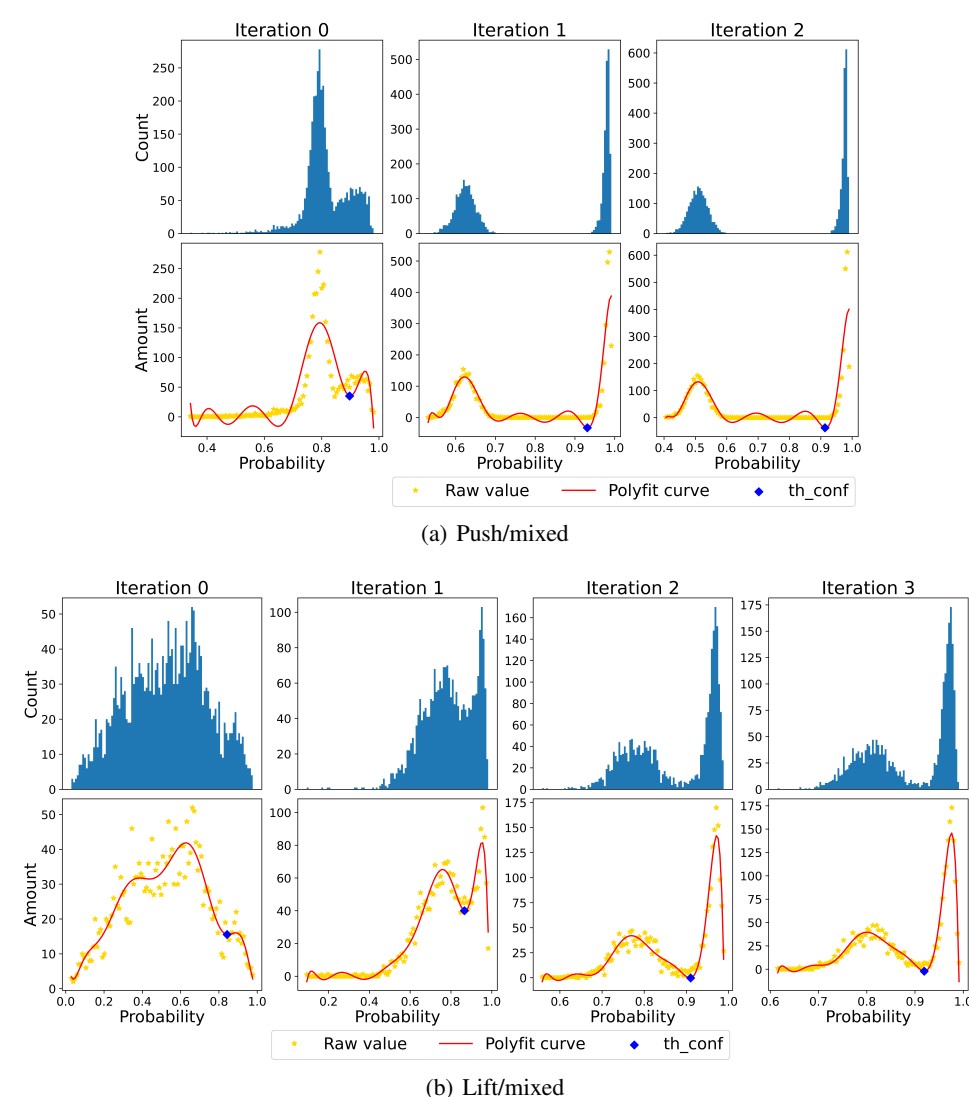

(a) Push/mixed

(b) Lift/mixed

Figure 3: A demonstration of selecting the adaptive *confidence* threshold, $th_{\mathrm{conf}}$, for the **Push/mixed** and **Lift/mixed** datasets. The top row of each subplot shows the confidence score counts for all trajectories in the larger dataset, $\mathcal{D}_{mix}$. The bottom row displays polynomial curves fitted to these histograms. The position of the blue dot indicates the rightmost local minimum of the polynomial, which determines the confidence value, $th_{\mathrm{conf}}$, used for the subsequent dataset filtering iteration.

## C Configurations used to collect the locomotion datasets

This section describes the acquisition process of the datasets used in Section 4. We selected two Mu-JoCo locomotion domains, namely *Ant-v3* and *Humanoid-v3*, which have relatively high dimensional state and action spaces, as well as high levels of difficulty.

Initially, we trained three different agents: target expert policy ($\mathbf{E}_+$); additional expert policy ($\mathbf{E}_-$); and weaker performing policy ($\mathbf{W}$). We utilized online RL algorithms for each domain. We trained each agent using different random seeds to ensure that the resulting agents can exhibit diverse behaviors or habits[3]. These trained agents were then deployed to interact with their corresponding environments. The interaction data was recorded in the form of $\{(s_t, a_t, r_t, y_t)_{t=1...n}\}$, where $y$ represents the ground truth label for each policy, indicating whether the data was collected by the target expert or not.

Table 4: Configuration details on the collection of MuJoCo datasets. We utilized the Soft Actor Critic (SAC) [51] and Twin Delayed Deep Deterministic Policy Gradient Algorithm (TD3) [52], as implemented in stable-baselines3 [53], using recommended hyperparameters. In the *Humanoid - v3* environment, the *reset_noise_scale* was set to $10^{-2}$.

| Subject | Ant | | | | Humanoid | | | |
|---|---|---|---|---|---|---|---|---|
| | $\mathbf{E_+}$ | $\mathbf{E_-}$ | **W** | **N** | $\mathbf{E_+}$ | $\mathbf{E_-}$ | **W** | **N** |
| Algorithm | TD3 | TD3 | TD3 | - | SAC | SAC | SAC | - |
| Train length | $10^6$ | $10^6$ | $2 \times 10^5$ | - | $10^6$ | $10^6$ | $3 \times 10^5$ | - |
| Mean return | 3034 | 2910 | 920 | - | 5725 | 5330 | 1576 | - |
| Data amount | $5 \times 10^5$ | $5 \times 10^5$ | $5 \times 10^5$ | $5 \times 10^5$ | $5 \times 10^5$ | $5 \times 10^5$ | $5 \times 10^5$ | $5 \times 10^5$ |
| Positive example amount | $2 \times 10^3$ | - | - | - | $2 \times 10^3$ | - | - | - |

In addition to these datasets, we also included noise data (**N**), in which the state, action, and reward were sampled using a uniform distribution within the range of minimum reward to maximum reward. Further details about the online training process, the performance of the trained agents, and the size of the collected datasets are provided in Table 4.

## D   Ablation study

In this section, we conduct an ablation study to examine the impact of various factors on the performance of our PU learning method. We have structured our study into several groups of ablations. Firstly, we aim to examine the techniques presented in Section 3.4. For each group in the study, we remove one technique to assess its effect on the overall performance. Additionally, we conduct an investigation into the size of the dataset of seed-positive dataset $\mathcal{D}_+$. In this part of the study, we gradually increase the size of the $\mathcal{D}_+$ starting from 0.1% to 1% of the unlabeled dataset $\mathcal{D}_{mix}$.

From the results shown in Table 5, it is easy to see that adaptive thresholds play a significant role in these relatively challenging physical robotic manipulation domains. Additionally, performing classification at the trajectory level can boost overall accuracy. The bagging is not the primary determinant of accuracy, but it can serve as a beneficial complement. Looking at the impact of the size of the positive dataset on PU learning, there were several direct failures when the dataset was extremely small. However, once the size increases to 0.4%, it is sufficient for PU learning to achieve optimal performance.

### D.1   Guidance on tuning the parameters

Based on our experience, an appropriately chosen set of hyperparameters can result in distinct dual peaks in the output histogram within as few as three iterations, as depicted in Figure.3. Among these peaks, the peak representing a higher probability corresponds to the expert data. Otherwise, it might indicate a set of poorly chosen hyperparameters; in which case, the following suggestions based on our experience may be helpful:

1. Setting an adaptive threshold is a critical technique that we highly recommend enabling, especially in scenarios where the expert policy and supposedly non-expert policy (or policies) used to generate the dataset exhibit similar behaviours. In adaptive thresholding, one crucial hyperparameter to consider is the order of the polynomial used to fit the histogram. If the order is set too high, it may lead to overfitting the histogram, causing the adaptive threshold to select thresholds at extremely high probabilities, which could fail to effectively distinguish a sufficient amount of expert data. On the other hand, if the order is set too low, it may miss the optimal threshold by not fitting the shape of the histogram well enough. We have found that a 10th-order polynomial yields the best results for our case. Empirically, when dealing with complex mixed datasets, such as our real-life-mixed dataset, we suggest slightly increasing the order to a range between 10 and 20. Conversely, in cases of simpler composite datasets, we recommend reducing the order to a range between 5 and 10.

2. If you observe that the histogram generated during training consistently exhibits a unimodal peak, this might be because you have not classified the trajectory by aggregating the state-actions within

Table 5: Results of the ablation study. Group 1 represents the removal of trajectory-based classification, instead opting for individual state-action pair classification. Group 2 employs a non-adaptive confidence threshold, using a fixed threshold set at 0.5 (considering the sigmoid interval of [0,1], the midpoint of 0.5 is arbitrarily chosen). Group 3 involves using a single classifier model instead of an ensemble (i.e., no bagging). Lastly, the 0.1% - 1% range refers to varying sizes of the $\mathcal{D}_+$, where the $\mathcal{D}_{mix}$ size is a fixed value. Examples of failure in classification are indicated by ✗, which signifies that the neural network's learning process has completely collapsed, rendering it unable to learn any meaningful information.

| Datasets | Ours | Group 1 | Group 2 | Group 3 | $\mathcal{D}_+$ size / $\mathcal{D}_{mix}$ size | | |
| | | | | | 0.1% | 0.4% | 1% |
| --- | --- | --- | --- | --- | --- | --- | --- |
| RRC-Real-**Lift/mixed** | 99.2% | ✗ | ✗ | 92.8% | ✗ | 99.2% | 99.2% |
| RRC-Real-**Push/mixed** | 100.0% | 77.8% | ✗ | 100.0% | 89.8% | 100.0% | 100.0% |
| Humanoid - **E+E** | 99.7% | 89.3% | 99.7% | 94.8% | ✗ | 99.7% | 99.7% |
| Humanoid - **E+W** | 100.0% | 94.8% | 100.0% | 100.0% | 69.7% | 100.0% | 100.0% |
| Humanoid - **E+N** | 100.0% | 98.7% | 100.0% | 100.0% | 100.0% | 100.0% | 100.0% |
| Humanoid - **E+E+W+N** | 99.0% | 83.2% | 99.0% | 91.0% | ✗ | 99.0% | 99.0% |

it. Classifying single state-action pairs may not effectively capture the inherent policy behavior. On the other hand, classification based on trajectories can take into account the correlation within the data to create a more precise classification relationship. This approach not only improves the model's accuracy but also accelerates the iterative process. Therefore, we recommend enabling this feature when applicable, such as in datasets that follow the D4RL protocol. Another potential reason might be that the initial amount of seed data you collected/separated was insufficient. This could prevent the neural network from effectively capturing the behavioral characteristics of the target expert data.

3. If you notice a converging trend in the number of expert trajectories being filtered out, but also observe substantial fluctuations (instability or lack of convergence) over subsequent iterations, you might consider increasing the number of classifier models in the ensemble. The quantity of models in the ensemble is a relatively straightforward hyperparameter that can be adjusted to ascertain the optimal value. This procedure is similar to the typical strategy of adjusting the learning rate.

# E    Benefits of PU learning in offline RL

This section aims to demonstrate the advantages of employing PU learning in stochastic offline RL algorithms. We utilize three medium-expert datasets available from the D4RL benchmark, specifically, *halfcheetah-medium-expert-v0*, *hopper-medium-expert-v0*, and *walker2d-medium-expert-v0*. These datasets have previously served as benchmarks in numerous studies [24, 26, 25] and have been effectively addressed by a range of algorithms. Our objective is to show that the application of PU learning can further enhance the performance of offline RL on these datasets.

The aforementioned datasets are all of mixed-quality, each of which was collected by two agents exhibiting different skill levels, specifically, medium and expert. Given that our approach requires a very small-scale seed-positive dataset, we extract the top 0.2% of trajectories based on cumulative reward from each mixed dataset to constitute the seed-positive data subset.

Our results, as displayed in Table 6, include comparisons with various state-of-the-art offline RL algorithms such as IQL [24], TD3+BC [26], and PLAS [25], with BC acting as the baseline. It is evident that the implementation of PU learning enhances agent performance. Examining the learning curves (shown in Figure 4), we can see that PU learning not only accelerates the learning process but also enhances the stability of all the investigated offline RL algorithms.

# F    Descriptions of compared algorithms

## F.1    PU learning

- **Unbiased PU learning** [42]: Traditional PU methods train classifiers by minimizing empirical risk, wherein unlabeled examples are directly treated as negative examples. This approach, however,

Table 6: Averaged normalized scores [15] with PU vs without PU for three D4RL benchmark tasks. Each result includes three random seed values and each model training session lasts for $10^6$ time steps. We evaluate each learned policy for 100 environmental trajectories.

|  | PU | BC | PLAS | IQL | TD3+BC | Total |
|---|---|---|---|---|---|---|
| Halfcheetah-medium-expert-v0 | ✓ | 0.93 | 0.92 | 0.93 | 0.94 | **3.72** |
|  | ✗ | 0.56 | 0.74 | 0.69 | 0.93 | 2.92 |
| Hopper-medium-expert-v0 | ✓ | 1.11 | 0.66 | 0.64 | 1.10 | **3.51** |
|  | ✗ | 0.51 | 0.32 | 0.35 | 0.89 | 2.07 |
| Walker2d-medium-expert-v0 | ✓ | 1.08 | 1.09 | 1.10 | 1.10 | **4.37** |
|  | ✗ | 0.76 | 0.99 | 1.06 | 1.11 | 3.92 |

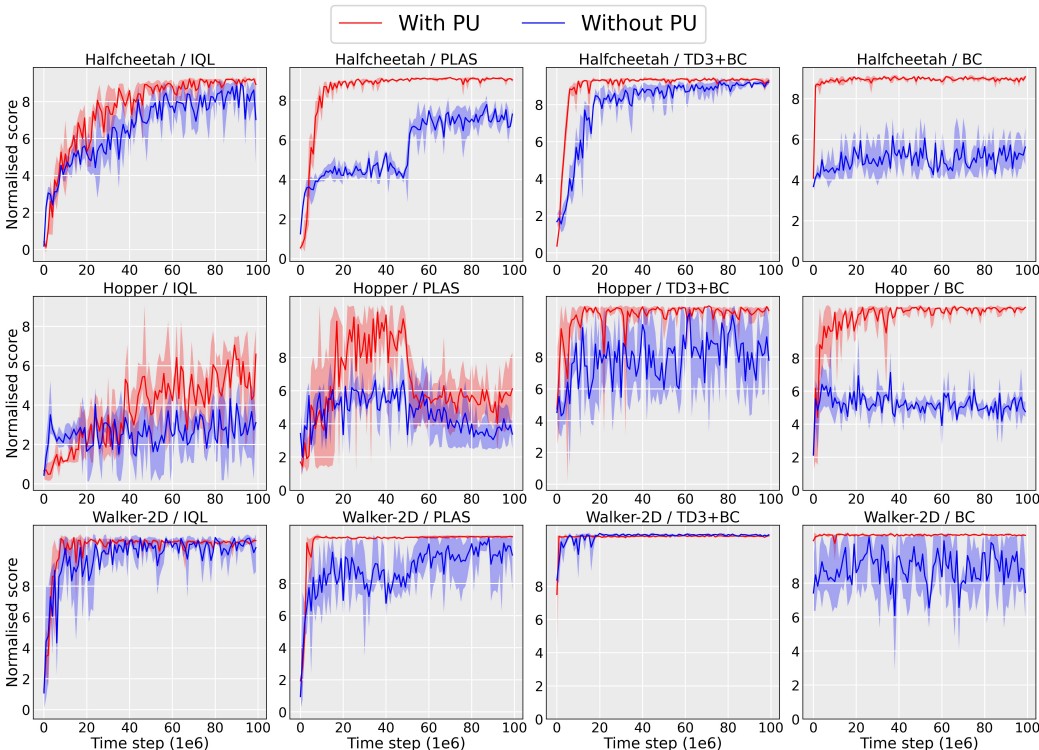

Figure 4: Training curves of Table 6; curves are averaged over three random seeds, with the shaded areas representing the minimum/maximum values across these three seeds. Each data point refers to the average normalised score [15] of 10 environmental episodes.

may lead to a high empirical risk for the negative class. To address this issue, Du Plessis et al. [42] re-weighted the losses for positive and unlabeled examples. Hence the training objective of Unbiased PU learning becomes minimizing the following, where $\delta$ is the proportion of positive examples to unlabeled examples in the unlabeled dataset (same as our approach, we set it to 0.4% here):

$$\mathcal{L}_{unbiased} = \delta \mathop{\mathbb{E}}_{(s_t,a_t)\sim\mathcal{D}_+}[-\log(\mathcal{F}(s_t,a_t))] + \mathop{\mathbb{E}}_{(s_t,a_t)\sim\mathcal{D}_-}[-\log(1-\mathcal{F}(s_t,a_t))] \\ -\delta \mathop{\mathbb{E}}_{(s_t,a_t)\sim\mathcal{D}_+}[-\log(1-\mathcal{F}(s_t,a_t))], \tag{3}$$

- **Non-negative PU learning**[43]: While as the complexity of the model increases, the risk (Equation 3) on the training set may approach or even become negative, while the corresponding risk on the test set increases. This suggests that the model is overfitting. To tackle this problem, Kiryo et al.

[43] introduced non-negative PU learning, in which the training objective is modified as follows, where again $\delta$ is the proportion of positive examples to unlabeled examples in the unlabeled dataset (same as our approach, we set it to 0.4% here):

$$
\mathcal{L}_{non-neg} = \delta \mathop{\mathbb{E}}_{(s_t,a_t)\sim\mathcal{D}_+} [-\log(\mathcal{F}(s_t,a_t))] + \max(0, \mathop{\mathbb{E}}_{(s_t,a_t)\sim\mathcal{D}_-} [-\log(1-\mathcal{F}(s_t,a_t))]
$$
$$
-\delta \mathop{\mathbb{E}}_{(s_t,a_t)\sim\mathcal{D}_+} [-\log(1-\mathcal{F}(s_t,a_t))]), \tag{4}
$$

Compared to Unbiased PU learning and Non-negative PU learning, the PU method introduced in this paper offers a more decent approach for generating negative examples, with improved diversity and logical correctness. Additionally, our method employs a simpler loss function.

### F.2 Policy learning algorithms

- **Naive reward-based filter + BC**: Reward may be used for filtering expert data for policy training, as experts are more likely to achieve higher rewards. Our results show two variations of this approach: 10%BC and 50%BC, which involve selecting trajectories with the top 10% and 50% highest cumulative (over the trajectory) returns, respectively, then using the filtered subsets to train BC.

- **Critic Regularized Regression (CRR)** [32]: A state-of-the-art filter/weight-based BC algorithm. It utilizes a reward-based advantage function to weight the significance of training examples in the dataset. In fact, the CRR method can be imagined as a soft-weighting process, while our method is a hard-weighting process.

- **Discriminator Weighted Behaviour Cloning (DWBC)** [38]: This is similar to our approch. DWBC employs a discriminator to distinguish high-quality data from mixed datasets. Additionally, it use a small seed-positive dataset to initiate the training process. The discriminator used here is similar to Unbiased PU learning, but they introduce the additional policy function as input. This approach combines the training process of BC with the discriminator to create a GAN-like architecture to improve performance. DWBC can be seen as a fusion of Unbiased PU learning and BC, whereas our approach introduces a novel PU learning method to guide BC.

- **Implicit Q-learning (IQL)** [24]: IQL is an offline reinforcement learning method that solves the OOD problem by setting implicit constraints. It has previously been used to address the problem of learning strategies in mixed-quality data.

- **Decision Transformer (DT)** [30]: In DT, a causal transformer is employed to model policy trajectories, taking as input the sequence $\{R_0, s_0, a_0, R_1, s_1, a_1, ..., R_{K-1}, s_{K-1}, a_{K-1}\}$, where $R$ represents the desired return-to-go in a trajectory; we specifically set this value to the maximum attainable reward for each domain. $s$ denotes the state and $a$ represents the action taken at each time step. At each time-step $t$, the first $3*t$ tokens are fed into the transformer to predict the action at time $t$, denoted as $p(a_t|R_0, s_0, a_0, ..., R_{t-1}, s_{t-1}, a_{t-1})$.

## G  Additional classification function

Here, we introduce an alternative option for the classification function discussed in Section 3.5. In this approach, we employ logistic operations to calculate the product of $\mathcal{F}$:

$$
f := \mathbb{1}\left[\sum_{k=1}^{K}\mathbb{1}\left[\left(\sum_{t=1}^{T}log(\mathcal{F}_k(s_t,a_t))\right) \geq th_{\text{conf}}\right] > \frac{K}{2}\right]. \tag{5}
$$

This would introduces a trade-off between disregarding positive data by being overly strict and incorporating non-positive data by being too lenient.

## H  Implementations and training

The neural network architecture for PU learning of the PUBC is illustrated in Figure 5. The PU training for the RRC Sim- and Real-**Lift/mixed** datasets lasted for 4 iterations; for the Sim- and

Table 7: Hyperparameters of algorithms used in our experiments

| Algorithm | Hyperparameter |
|---|---|
| PU | learning_rate=0.001; batch_size=1024; optimizer = adam; epochs_per_iteration=20; models_in_ensemble=3; polynomial_order=10 |
| BC | learning_rate=0.001; batch_size=100; optimizer = adam; epochs=200 |
| CRR | actor_learning = critic_learning_rate = 0.0003; batch_size=256; optimizer = adam; beta=1.0 |
| IQL | actor_learning_rate=critic_learning_rate=0.0003; batch_size=256; optimizer = adam; expectile=0.7; weight_temp=3.0 |
| DWBC | learning_rate=0.0001; batch_size=256; alpha=7.5; eta=0.5; no_pu=False |
| TD3+BC | actor_learning = critic_learning_rate = 0.0003; batch_size=256; alpha=2.5 |
| PLAS | actor_learning_rate=0.0001; critic_learning_rate=0.001; optimizer = adam; warmup_steps=500000; beta=0.5 |

Real-**Push/mixed** datasets, 3 iterations; and for each MoJoCo dataset, 3 iterations. In each iteration, the subset of negative examples, $D_-$, is set to have the same size as the subset of positive examples, $D_+$. Furthermore, the number of each type of negative example, $(n_1 - n_7)$, remains consistent throughout the subset of negative examples, $D_-$.

Once our work is accepted, we plan to open-source our full implementations of PUBC on GitHub. The implementations of the baseline PU learning algorithms from `https://github.com/cimeister/pu-learning`. All the offline RL and BC algorithms used in our study are sourced from d3rlpy [54]. The implementation of DWBC [38] is based on the original authors' work at `https://github.com/ryanxhr/DWBC`.

The key hyperparameters for training each algorithm involved in our experiment are displayed in Table 7. Our experiments ran on a PC with an Intel I9-12900F CPU (2.40 GHz $\times$ 16, 32 GB RAM) and an NVIDIA 3090 GPU.

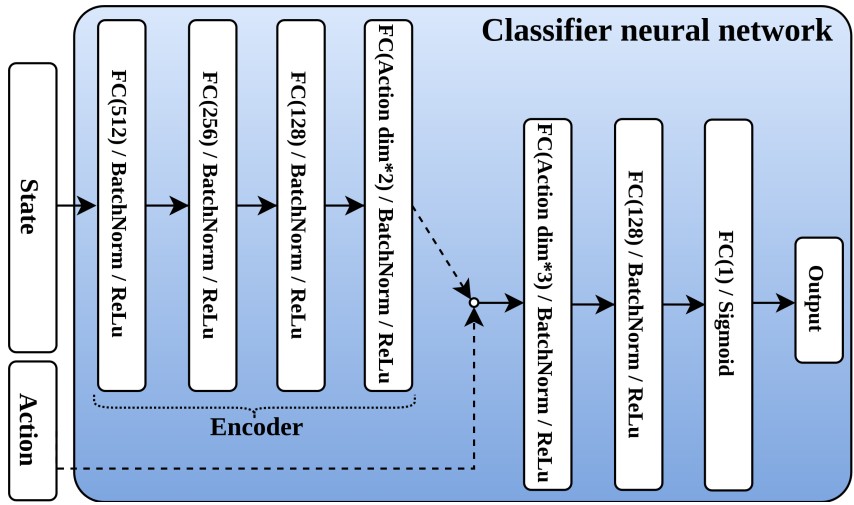

Figure 5: The neural network structure of classifier, illustrating the details of Figure 1(b).

