# OpenReview forum: "Improving Behavioural Cloning with Positive Unlabeled Learning"
_robot-learning.org/CoRL/2023/Conference — CoRL 2023 Poster_

### Official Review · Reviewer_oaHQ · 2023-07-04

**Confidence:** 3
**Originality:** Good
**Technical Quality:** Very Good
**Clarity Of Presentation:** Very Good
**Impact:** 3

**Recommendation:**

Strong Accept: I recommend accepting the paper and will argue for my recommendation even if other reviewers hold a different opinion.

**Review:**

# Strengths and Weaknesses

## Strengths

******************Writing:****************** Overall the paper is pretty well-written an easy to understand. Section C in the appendix does a good job demonstrating how the confidence threshold selection process actually makes quite a bit of sense.

****************Method:**************** The PUBC method is quite simple and demonstrates good performance across a moderately comprehensive set of environments. The authors make some very reasonable design choices, such as thresholding entire trajectories rather than individual (s, a) tuples.

**************************Experiments:************************** Experiments are pretty thorough if including the appendix experiments (the one demonstrating PUBC helps with Offline RL is nice).

## Weaknesses

************************Motivation:************************ The motivation for this paper as described in the intro isn’t that convincing (L 40-42). The paper focuses on the setting where a mixed-quality dataset is provided from which we are trying to extract a single expert’s data from. But why would the expert not label their own data? That is the simplest thing to label when collecting data. A slightly better motivation is presented in L46-48, stating that the authors are attempting to extract data that shares the same patterns as the *************seed-positive************* provided dataset. Perhaps the best way to frame this paper is with respect to *********reward annotations:********* It can be very expensive to annotate rewards for all transitions, especially in complex, real-world settings, so therefore we can use PUBC to extract high-performing trajectories from a mixed quality dataset *************************by providing a few demos************************* instead of manually creating a reward function.

************************Experiments:************************

- Section 4.2 has no *context* for *why* the authors’ method works better than the unbiased and non-negative PU learning baselines. The authors should put a short description of each and their differences for context. (on the other hand 4.3 does a good job)

- I do think more simulated environments should be included in the main paper for comparison with offline RL (why not the entire D4RL suite?), given that many offline learning paper do compare on many mujoco environments.

**************************Minor issues:**************************

- L268: “Is not group” → “is not grouped”

**Quality Of The Limitations Section:**

Limitations are addressed clearly

**Questions For Rebuttal:**

Please see weaknesses above for details.

I like the method but I think the motivation could be better presented throughout the paper and especially in the introduction.

**Robotics Focus:**

Sufficient demonstration on hardware

**Summary Of Paper:**

Proposes a semi-supervised method (”Positive unlabeled”) which is able to use a seed dataset of expert trajectories to determine what trajectories are best for training on for BC from a large set of mixed-quality datasets, without the need for reward annotations.

**Summary Of Recommendation:**

The paper has sufficiently comprehensive experiments including on real robots and the method is intuitive and performant across a variety of metrics.

I would recommend more environments and perhaps a better-presented motivation, but to me this is still at least a weak accept at the moment.

---

### Official Review · Reviewer_aoto · 2023-07-08

**Confidence:** 4
**Originality:** Good
**Technical Quality:** Very Good
**Clarity Of Presentation:** Very Good
**Impact:** 3

**Recommendation:**

Strong Accept: I recommend accepting the paper and will argue for my recommendation even if other reviewers hold a different opinion.

**Review:**

Strengths
- The mixed-quality dataset was mostly utilized while maintaining imitation in an offline manner.
- The proposed method achieved the solid performance improvement through extensive validation, including on real robot.

Weaknesses
- The convergence of the process of extracting data from D_mix to D_+ is not given, so the worst-case computation time for learning is unclear.
- It is difficult to distinguish between trajectories included in D_+ and trajectories that yield large returns (should we define some meta objectives?), so how to construct D_+ in practice is unclear.


**Quality Of The Limitations Section:**

Additional details required

**Questions For Rebuttal:**

While the proposed discriminator can evaluate whether a state-action pair belongs to D_+ or not at each time step, the proposed method classifies whether a trajectory should be moved from D_mix to D_+.
It is understandable that the judgment at each time step is prone to misjudgment and does not contribute to performance improvement as shown in the Ablation study, but there must be some trajectories in D_mix that are partially optimal but fail in the rest.
If this is the case, then it would be considerable to classify whether adding at each segment of trajectories with appropriate horizon since it might avoid misjudgments enough and ignore non-positive data contained in the trajectories.

Also, the classification per trajectory is calculated by the summation (average) of F, but the product of F would be also valid.
I think this is a tradeoff between ignoring positive data by judging too strictly and adding non-positive data by judging too loosely, so I would like to see the discussion which is more effective.


**Robotics Focus:**

Sufficient demonstration on hardware

**Summary Of Paper:**

In this paper, the authors proposed a method to improve the imitation performance of offline imitation learning with a mixed-quality dataset in addition to a positive dataset by classifying and extracting trajectories similar to ones in the positive dataset, which were actively used for training BC.
For this purpose, the authors developed an algorithm that learns the discriminator whether a given state-action pair should be in the positive dataset or not by generating negative examples heuristically.
As a result, the authors contributed to a solid improvement, including a first place in a competition.


**Summary Of Recommendation:**

The effective use of the mixed-quality dataset proposed in this paper is well done and is worthy of publication, as it shows a definite improvement in performance.
However, the criteria for deciding whether or not to add data from D_mix to D_+ are still open to debate.

---

### Official Review · Reviewer_KCwR · 2023-07-12

**Confidence:** 4
**Originality:** Good
**Technical Quality:** Very Good
**Clarity Of Presentation:** Very Good
**Impact:** 3

**Recommendation:**

Weak Accept: I recommend accepting the paper, but will not argue for my recommendation if the majority of other reviewers have a different opinion.

**Review:**

**Strengths**
* This is a simple and well executed idea and will be a helpful baseline for future offline RL papers.
* The methods are convincingly tested in real world using the real robot challenge [4]

**Weaknesses**
* The most obvious missing baseline here seems to be other (potentially more scalable?) alternatives for reward filtered imitation learning via sequence modelling -- like Decision Transformer [5] and Trajectory Transformer [6]. The authors should make a paragraph for this in their related work section, and (potentially) add these methods as baselines.
* The PUBC method seems to have a lot of moving parts (e.g. ensemble, thresholding, etc.) to generate a score for each transition. An ablation study would be helpful to understand what parts of the proposed classifier scheme are most important.


[4] R. McCarthy, F. R. Sanchez, Q. Wang, D. C. Bulens, K. McGuinness, N. O’Connor, and S. J Redmond. Solving the Real Robot Challenge using deep reinforcement learning.

[5] Chen, L., et al. Decision transformer: Reinforcement learning via sequence modeling.

[6] Janner, M., Li, Q., & Levine, S. Offline reinforcement learning as one big sequence modeling problem.

**Quality Of The Limitations Section:**

Limitations are addressed clearly

**Questions For Rebuttal:**

Please address the weaknesses I listed above.

**Robotics Focus:**

Sufficient demonstration on hardware

**Summary Of Paper:**

This paper proposes a positive-unlabeled learning scheme for implementing filtered Behavior Cloning. Given a handful of successful demonstrations and a large replay buffer of mixed quality (e.g. some optimal transitions, many sub-optimal), this algorithm first learns an ensemble of classifiers (w/ positive unlabeled learning) to separate out the good transitions in the buffer. Then an agent is trained on these optimal transitions (original optimal demos + filtered transitions from buffer) using behavior cloning.

The proposed method shows a significant improvement over simple/obvious reward filtered BC baseline, and also improves over competing offline RL algorithms (IQL[1] /DWBC [2] /CRR [3]).

[1] I. Kostrikov, A. Nair, and S. Levine. Offline reinforcement learning with implicit q-learning.

[2] H. Xu, X. Zhan, H. Yin, and H. Qin. Discriminator-weighted offline imitation learning from suboptimal demonstrations.

[3] Z. Wang, et al. Critic regularized regression.

**Summary Of Recommendation:**

This is a good, well-executed paper that will (at the very least) serve as a good "sanity check" baseline for future offline-RL work. The only important issue is missing discussion of reward + sequence modelling [5,6] in related work.

---

### Official Review · Reviewer_MudH · 2023-07-19

**Confidence:** 4
**Originality:** Good
**Technical Quality:** Good
**Clarity Of Presentation:** Good
**Impact:** 3

**Recommendation:**

Weak Accept: I recommend accepting the paper, but will not argue for my recommendation if the majority of other reviewers have a different opinion.

**Review:**

Strengths
- A well-executed pipeline with a combination of simple or already existing methods, exhibiting the strong experimental results. Not every paper should have fancy novel components and I appreciate the main contribution of this paper.
- Clear writing with illustrative figures and pseudocode

Weaknesses
- Qualitative analysis that illustrates how the illustrative labelling procedurue continues can be helpful for understanding the method, like providing the learning dynamics with accuracy or some visualization can be helpful.
- It seems like the proposed pipeline quite heavily depends on the additional methods proposed in Section 3.4, as the ablation study shows that the method collapses without them. It might be worth moving the ablation study results to the main section and put a more discussion and practical guides for future researches, which could be helpful for fully understanding how the method works and useful for researchers who are willing to adopt the pipeline.
- 'Accuracy' in the caption of Table 1 is a bit confusing. Please specify what 'accuracy' means here.
- Please fix minor typos in the draft (e.g., Perfroms in line 85, Section ?? in line 410)

**Quality Of The Limitations Section:**

Limitations are addressed clearly

**Questions For Rebuttal:**

Please address the concerns and questions in Weaknesses.

**Robotics Focus:**

Sufficient demonstration on hardware

**Summary Of Paper:**

This paper presents a pipeline for improving the behavior cloning from mixed quality datasets. The main idea is to adopt the idea of semi-supervised learning that iteratively labels the samples as positive when the classifier is certain about its prediction. Specifically, it learns the classifier on a small set of expert demonstrations and then iteratively labels the remaining dataset. And the paper introduces several techniques that can further enhance the pseudo-labelling procedure. The method is evaluated on both real-world and simulated benchmarks.

**Summary Of Recommendation:**

This paper presents a pipeline with a combination of well-executed techniques, exhibiting strong experimental results on both real-world and simulation benchmarks. But it's missing some qualitative analysis and detailed discussion on several components crucial for making the method indeed work, hence I recommend the score to be weak accept at the current status of the draft.

---

### Decision · Program_Chairs · 2023-08-30

**Decision:**

Accept (Poster)

**Comment:**

The reviewers have agreed that the paper presents a conceptually simple yet effective method to an important problem (offline RL) with extensive experimental evaluation. The AC agrees with the reviewer's assessment and recommend the paper to be **accepted** and presented as a **poster**.